# Early-Age Tensile Bond Characteristics of Epoxy Coatings for Underwater Applications

**Sungwon Kim [1], Hyemin Hong [1], Taek Hee Han [2] and Min Ook Kim [1],***

[1]  Coastal Development and Ocean Energy Research Center, Korea Institute of Ocean Science and Technology, 385 Haeyang-ro, Yeongdo-gu, Busan 49111, Korea; swkim@kiost.ac.kr (S.K.); hyeminhong@kiost.ac.kr (H.H.)
[2]  Research Project Development Affairs Section, Korea Institute of Ocean Science and Technology, 385 Haeyang-ro, Yeongdo-gu, Busan 49111, Korea; taekheehan@kiost.ac.kr
*  Correspondence: minookkim@kiost.ac.kr

**Abstract:** In this study, coating equipment for the effective underwater repair of submerged structures was developed. The tensile bond characteristics of selected epoxy resin coatings were investigated by coating the surface of a specimen using each of the four types of equipment. Using the experimental results, the tensile bond strength and the coating thickness were analyzed according to the type of equipment, coating, and curing time. The results show that the type of coating equipment used had the greatest effect on the measured bond strength and coating thickness of the selected coatings. However, the effect of coating type and curing time on the bond strength and the thickness was observed to be insignificant. Compared with the developed equipment, the surface treatment of the coating was observed to be more effective when using the pre-existing equipment, and thus the bond performance of the coating was improved compared to using the pre-existing equipment. Based on the experimental results, improvements and needs involving the equipment for further research were discussed.

**Keywords:** underwater application; epoxy coating; tensile bond characteristics; coating equipment; marine structures

## 1. Introduction

In ports and oceans, there are a large number of structures used for various purposes such as ship anchoring, leisure activities, and oil drilling. These structures are designed for a service life of more than 50 years for port structures and 20 to 25 years for fixed offshore structures [1,2]. Marine and coastal structures are exposed to harsh environmental conditions such as waves and tides that damage the structures through corrosion, deterioration, and weathering [3–13]. The sizes of offshore structures are very large and are likely to undergo severe damage during an accident. Moreover, the damage can be accompanied by casualties, meaning the maintenance of the structure is important. Various materials such as steel, concrete, and fiber-reinforced polymer (FRP) are utilized for constructing marine and offshore structures, and one of the typical maintenance methods for structures constructed using these materials involves using underwater-coating technology. This technology involves the application of the coating material on the surface of the underwater structure to prevent, in advance, the penetration of harmful substances into the structure and extend its life. Ferrari et al. [14] conducted research on superhydrophobic coatings using recyclable materials. Separate research related to the corrosion prevention of steel in seawater using passive films was also conducted [15]. Kim et al. [16] suggested that the simplest method of reinforcing underwater structures was to coat their surface using a coating material that is workable underwater. They also stated that developing an epoxy material which is workable underwater was the most effective way of applying this method. As a

result, they developed two highly efficient materials, namely, the aqua repair adhesive (APA) and the aqua impregnation epoxy (AIE). The optimum mixing ratio between base resin and hardener was also derived, and the performance of APA and AIE was verified using the pull-off strength test, bond shear strength test, and chemical resistance test. Furthermore, APA and AIE did not dissolve well in water, produced no residual particles during underwater curing, and exhibited almost the same adhesion as that in dry conditions.

Galliano et al. [17] developed an epoxy coating for the corrosion protection of steel and analyzed its characteristics. In addition, many studies on underwater coating materials have been conducted [18–21], and some epoxy coatings have been commercialized. However, only few research studies have been conducted on the underwater-coating equipment, and the existing commercialized coating materials are directly applied to underwater structures using either a general paintbrush or a worker's hand. In Idaho National Engineering and Environmental Laboratory, experiments were performed using a method of squeezing a coating material with a caulking gun [22]. However, the method was time consuming as each squeeze provided just a small quantity of coating material, and the material also did not mix well.

If the coating material is applied on the surface of the submerged structure without using coating equipment, then there is heavy loss of the coating material owing to underwater environmental factors such as the water current caused due to moving the coating-material-storage container to the brush or due to the operator's hand. Moreover, the diffused underwater coating materials are more likely to adhere to the diving equipment (e.g., masks, respirators, and oxygen tanks), thereby threatening the operator's safety. This study aims to develop underwater-coating equipment that can reduce the material loss, ensure the safety of workers, and enable efficient underwater repair work. The two developed equipment were examined and their bond performances were compared with those of existing coating methods. Commercially available epoxy coatings were carefully selected for testing the equipment, and their tensile bond characteristics were analyzed.

## 2. Experimental Investigation

Four different types of coating equipment (two existing and two developed) were tested using six different commercially available epoxy coatings. The experiment was performed indoors, and the coatings were applied on the surface of specimens in underwater condition. The applied coatings were subjected to three different durations (24, 48, and 72 h) of curing time, following which the thickness and tensile bond strength of the coatings were measured to verify the performance of the coating equipment and to analyze the bond properties. The details of the coating equipment and the experimental procedure are described below.

### 2.1. Coating Equipment for Underwater Application

In the experiments conducted in this study, two existing types of equipment and two developed equipment were used for applying coatings. The conventional paint roller (EQ1) used for general paintwork, and the pressure roller (EQ2) which is used via an airless pump, were selected as the existing equipment. As shown in Figure 1, EQ2 connects an airless pump unit to a pressure roller to keep supplying the coating material using the concept of air-pressure difference. However, when using EQ1, the operator needs to dip the roller directly into the coating material to replenish the coating material while EQ2 can use the pressure of the airless pump to supply the coating material to the pressure roller, thereby reducing the coating material loss underwater. The maximum operating pressure of the airless pump supplying the coating material is 3300 psi, and the frame of the pressure roller is connected to the pump using a hose [23]. The pressure-controlling trigger is mounted on the frame part such that the rate of filling the coating material into the pressure roller can be adjusted.

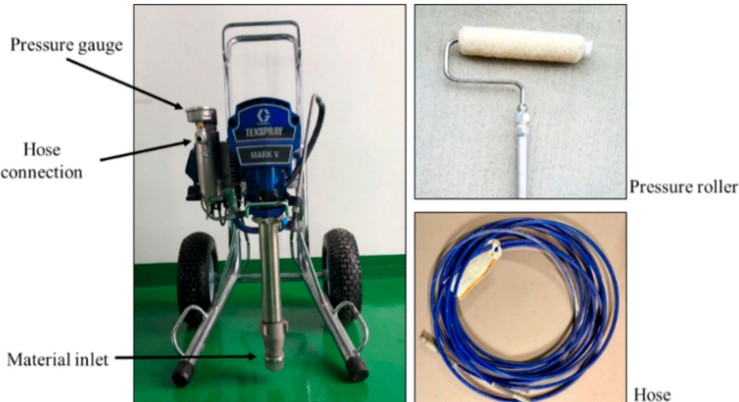

**Figure 1.** Selected equipment (EQ2) that using airless pump.

### 2.1.1. Equipment Using Air Pressure (EQ3)

Figure 2 shows the coating equipment (EQ3) developed in this study that uses air pressure. The coating stored in the tank ejects using the pressure of the air stored in the other tank. Both the tanks were manufactured using 3.0 mm stainless steel 10S designed to withstand the maximum internal pressure of 20 kg/cm$^2$. Moreover, to improve the airtightness of the tanks, tungsten inert gas welding was performed, and the joints were connected by both general welding and reinforcing welding. Valves were also installed for controlling the air pressure and discharge rate of the coating material. In addition, a pressure gauge was employed to check the remaining air pressure in the air tank, such that additional air could be timely supplied to the air tank using an air compressor to avoid insufficient air pressure during the test. The diameter of each of the nozzle and the compressed air supply line was selected considering the high viscosity and density of the coating material. The coating material moved to the outlet using air pressure and was finally jetted out radially from the spray nozzle connected to the outlet (see Figure 3). In the design stage, using the CFD analysis, it was confirmed that the spraying of the coating material was performed radially, as depicted in Figure 3. Finally, a brush was attached at the end of the spray nozzle to conveniently apply the coating material on the intended surfaces. Each part of the equipment, including the brush, are composed of a material that is resistant to both the corrosion and the chemical modification due to the thinner used for cleaning. Furthermore, all the parts except the two tanks, handles, and pedestal can be disassembled. To improve workability, the parts were connected using clamps for easy disassembly or assembly.

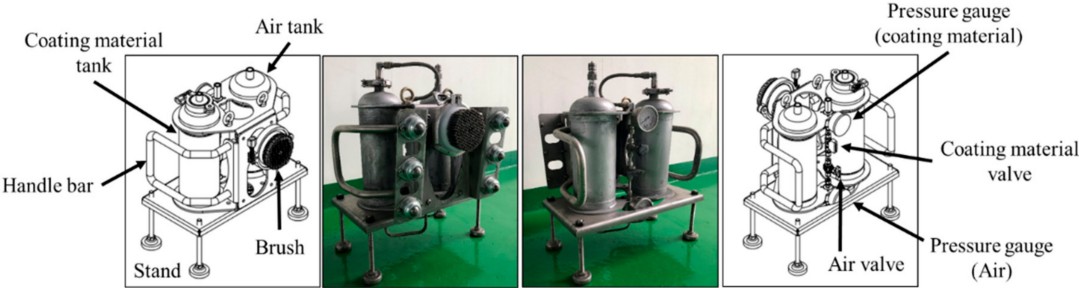

**Figure 2.** Front and rear parts of EQ3.

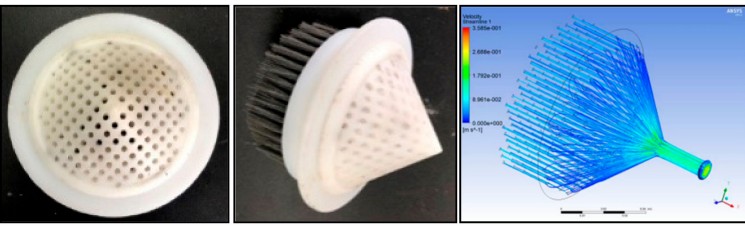

**Figure 3.** Nozzle and brush for EQ3.

### 2.1.2. Equipment Using Spring Stiffness (EQ4)

The equipment EQ4, depicted in Figure 4, was developed and inspired from a silicone gun. EQ4 pushes the coating material in the container using a piston that operates under spring stiffness. The body of EQ4 has two inlets, one for filling the coating material and the other for regulating its internal pressure. Opening the latter inlet adjusts the internal pressure of the body, thereby thrusting the piston out to make space for the coating material to be filled into the body. Subsequently, the coating material is filled into the body through the former inlet. Finally, by closing both the inlets, the equipment becomes ready for jetting out the coating material. Upon opening the valve while the coating material is completely filled therein, the coating material is jetted by the piston's movement in the nozzle direction because of the spring stiffness. Furthermore, a nozzle is connected to the end of the outlet of the coating material, as in the case of EQ3, and the coating material discharges radially. Velcro is attached to the end of the nozzle to make it easy to attach and detach from a wool brush, which is used for surface treatment. The surface of the specimen with worn out coating material is treated under a certain pressure by the operator using the wool brush depicted in Figure 5.

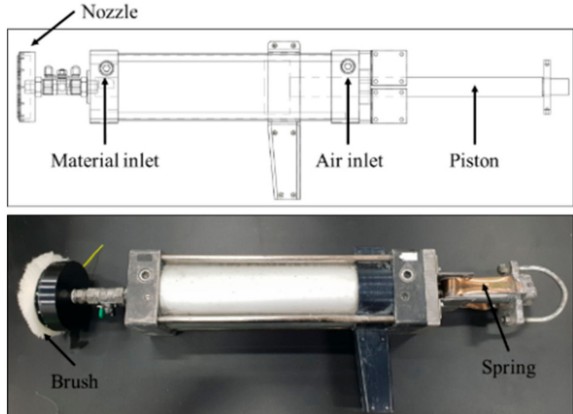

**Figure 4.** EQ4-Gun type.

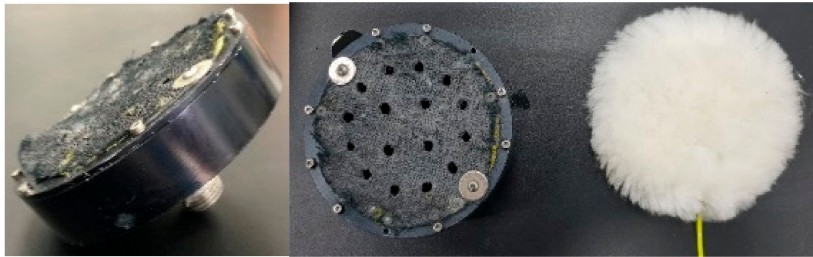

**Figure 5.** EQ4-nozzle and wool brush.

### 2.2. Coating Materials

The underwater coating materials used were selected from six commercially available epoxy-based coating materials. Each coating material is a two-component system using a mixture of the main resin

and the hardener. The mixing ratio of the main resin and the hardener was tested according to the manufacturer's recommendation. The properties of the selected coating materials are listed in Table 1.

**Table 1.** Material properties of selected coatings.

| Coating Material (Code) | Tensile Bond Strength (MPa) | Density (g/cm$^3$) | Working Time (min) | Product Name | Company |
|---|---|---|---|---|---|
| S1 | 12.7 | 1.75 | 45 @ 20 °C | 5831 | Belzona |
| S2 | 16.6 | 1.60 ± 0.1 | 30 @ 30 °C | RS500P | Chemco |
| S3 | 12.0 | 1.6 | 30 @ 25 °C | M500 | Westsea |
| S4 | 6.9 | 1.55 | 45–60 @ 20 °C | 28.15 | A&E Systems |
| S5 | 17.0 | 1.82 | 45–60 @ 20 °C | 28.14 | A&E Systems |
| S6 | Not available | | 40 @ 23 °C | 5027 | AkzoNobel |

It was observed that the working time (pot life) of the coating material used in the experiment was closely related to the working temperature (see Figure 6). Notably, the pot life of a coating material plays a significant role because it becomes difficult to continue the underwater coating work as the curing time of the coating material begins immediately after the pot life. The experiment was conducted under the environmental conditions of 13.5–18.5 °C and proceeded stably.

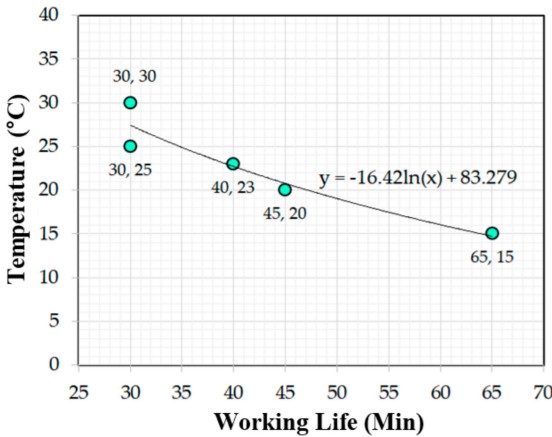

**Figure 6.** Relationship between available working time and outside temperature.

*2.3. Test Procedure and Measurement*

For testing the performance of the developed underwater-coating equipment and analyzing the characteristics of tensile bond strength of the selected underwater-coating material, the experiment was performed inside a circular water tank filled with fresh water. The experimental process is depicted in Figure 7. Six coating materials were applied on the surface of the specimen using four different types of coating equipment. The steel grade of the specimen was SS400, and the size was 300 mm × 300 mm × 3 mm. The coating material was applied on the surface of the specimen in the underwater condition, following which the specimen was stored in a separately prepared curing tank to allow the coating material to cure for the predetermined curing time. The coating material was applied to three specimens for every 24 cases (six materials and four types of equipment), and the specimens were then stored in the curing tank for 24, 48, and 72 h. Subsequently, the specimens were taken out on the land, and the tensile bond strength of the coating material was analyzed according to the curing time.

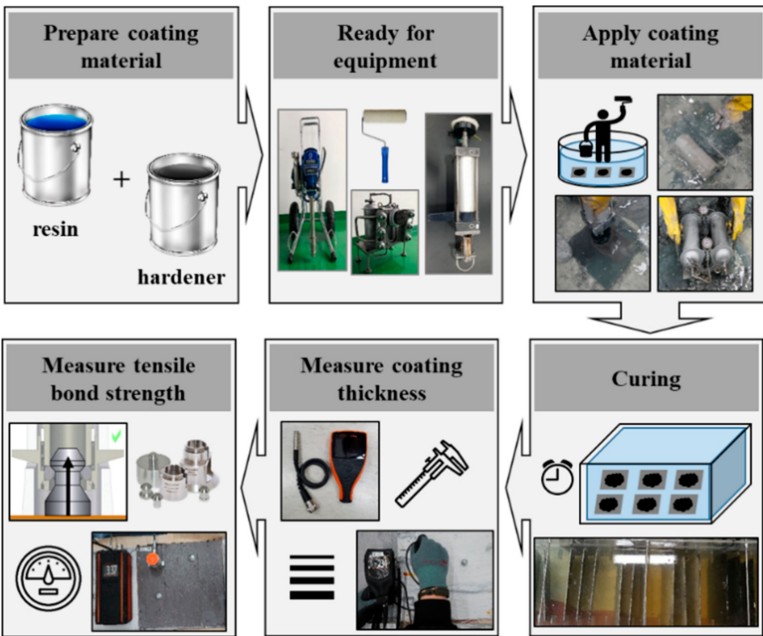

**Figure 7.** Detailed coating procedure.

All the experiments were performed in underwater conditions, and it would have been ideal if the measurements for both bond strength and thickness were also performed in the underwater condition. However, the rules or criteria for measuring the thickness and tensile bond strength of the coating film in underwater condition have not been determined yet; therefore, the measurements were performed according to the method on land, as depicted in Figure 8. After the curing time had elapsed, the specimens were sequentially taken out of the curing tank to measure the thickness and tensile bond strength of the applied coating material. The water on the surface of the coated material was removed using an experimental wiper, following which the surface was left for approximately 10 min to dry completely. Subsequently, the tensile bond strength and the thickness of the coating material were measured according to the standard method depicted in Figure 9.

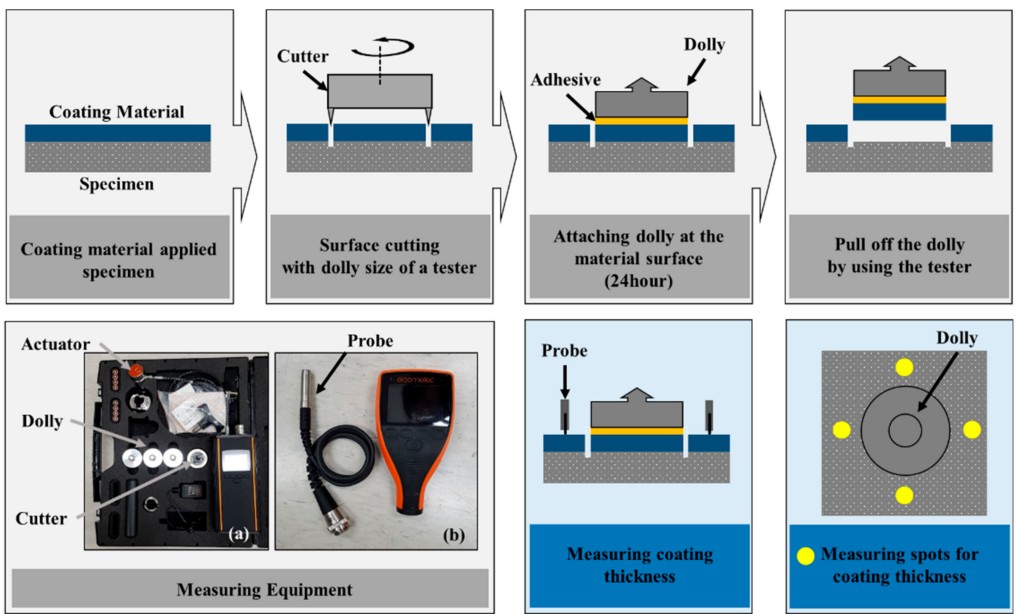

**Figure 8.** Measurements of pull-off bond strength and coating thickness. (**a**) pull-off adhesion gauge; (**b**) coating thickness gauge.

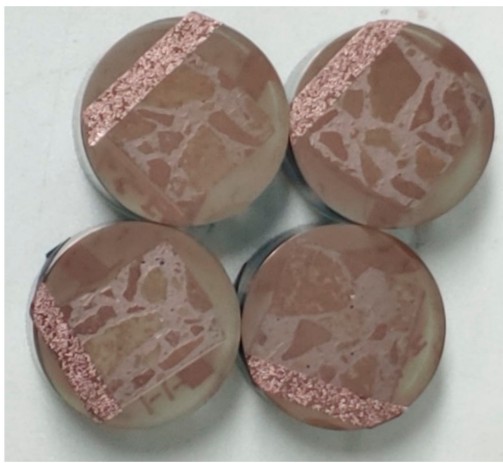

**Figure 9.** Prepared samples for SEM analysis.

The equipment, Elcometer F510-50S (Elcometer, Manchester, UK) [24], for measuring the tensile bond strength is depicted in Figure 8a. The surface of the specimen was visually checked to determine points for the thickness measurement and the dollies were attached to the surface using an adhesive. The dolly-attached specimens were then stored on a working table for 24 h to ensure sufficient curing of the adhesive. Subsequently, each dolly was pulled out at the rate of 0.10 MPa/s to measure the tensile bond strength of the coating material. As depicted in Figure 8b, coating thickness was measured using an Elcometer A456CFBI1 (Elcometer, Manchester, UK) [25] that measure the thickness using magnetic properties. Notably, the thickness was measured prior to removing the dolly from the four points at the top, bottom, left, and right around the attached dolly, as depicted in Figure 8.

*2.4. Sample Preparation for SEM Analysis*

Scanning electron microscope (SEM) analysis was carried out to investigate the interfacial bond between applied coating and substrate. For preparing the sample for SEM, the coated specimens were cut with a high speed saw. The sample size is $20 \times 20 \times 20$ mm$^3$. The cold mounting work was performed using the adhesive composed with epoxy resin and hardener (mixed in the weight ratio 100:12). The samples were cured for 8 h within a 32 mm diameter mounting cup and grinding was conducted for 5 min with using 600 grit abrasive. A total of four stages of polishing were carried out, namely, 9 μm abrasive for 10 min, 3 μm abrasive for 5 min, 1 μm abrasive for 5 min, and finally 0.05 μm abrasive for 3 min. SEM images were taken at magnification of 1000× using EM-30AX manufactured by COXEM, Daejeon, Republic of Korea. Figure 9 shows the prepared samples for SEM analysis.

**3. Results and Discussion**

Test results were summarized based on measured tensile bond strength and coating thickness values according to each test variable (coating equipment, coating material, and curing time). Figure 10 depicts the six coating materials applied on the surface of the specimens using an ordinary roller (EQ1). The specimens were cured underwater for 72 h, and three dollies were attached to each specimen.

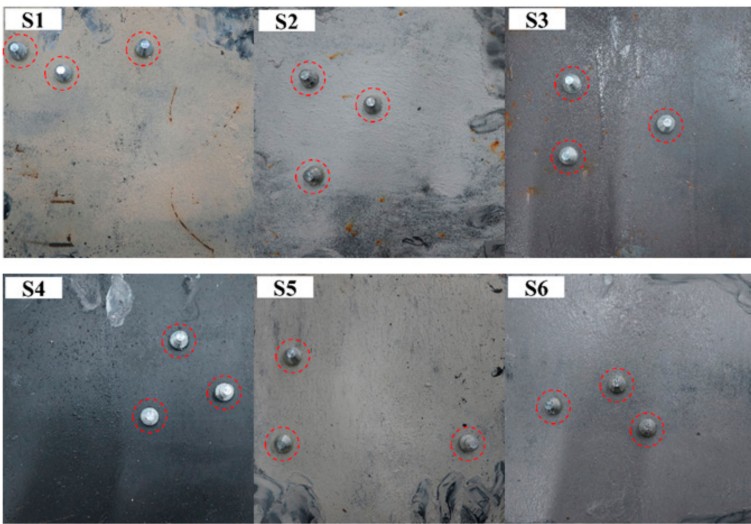

**Figure 10.** Coated specimens after 72 h water curing.

*3.1. Effects of Coating Equipment on Measured Bond Strength and Coating Thickness*

The effects of coating equipment on the tensile bond strength and the coating thickness are depicted in Figure 11. Upon applying the coating material using EQ1, EQ2, EQ3, and EQ4, the average/standard deviations (SDs) of tensile bond strengths were 3.60 MPa/2.42, 1.46 MPa/0.57, 2.70 MPa/1.42, and 2.83 MPa/1.51, respectively. Furthermore, the highest tensile bond strength was exhibited for the case using EQ1. The highest value of the average film thickness was 0.38 mm (with SD 0.07), for EQ3. Upon using the equipment EQ1, EQ2, and EQ4, the averaged coating thicknesses were 0.08 mm (with SD 0.13), 0.14 mm (with SD 0.26), and 0.04 mm (with SD 0.02), respectively.

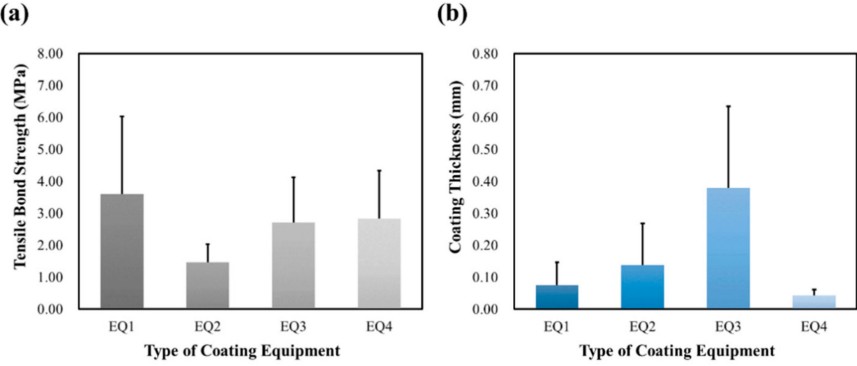

**Figure 11.** Effects of coating equipment on measured tensile bond strength and coating thickness at 72 h curing ((**a**) bond strength; (**b**) coating thickness).

It was confirmed that the effect of the equipment type on the measured tensile bond strength and the coating thickness values was significant. In more detail, EQ1 performed the best in terms of tensile bond strength, while EQ4 performed the best in terms of coating thickness and uniformity. Furthermore, in the case of the developed equipment, jetting of the coating material was smooth, but the surface treatment was different from those of existing equipment. In the case of the pre-existing equipment, EQ1 and EQ2, the rolling surface-treatment method was used. However, in the case of the developed equipment, EQ3 and EQ4, the brushing surface-treatment method was used. Upon applying the coating material using EQ1, which was the lightest equipment, the sensation of application of the coating material could be directly felt by the operator. However, when using other equipment, the operator experienced difficulty in feeling a similar sensation. In particular, it was more difficult to treat the surface by brushing in the case of EQ3 and EQ4, as the mass of each of these was much higher than that of EQ1. Moreover, in the case of EQ3, compared with the other equipment, high-density

stiff material brushes were used, thereby rendering the surface treatment more difficult. As can be seen from the above-mentioned results, the thickness of the coating film was the largest and the SD was very large for EQ3, indicating that the surface treatment was not stable. EQ4 delivered the best surface-treatment results owing to the use of the softest wool brush despite the difficulty of working due to EQ4's large mass. On the other hand, the high viscosity of the coating material caused small amounts of the wool brush to fall off, thereby limiting the surface treatment of a large area. To improve work convenience and surface treatment, it is necessary to compensate for disadvantages such as bulkiness and unsatisfactory brush performance of the developed coating equipment.

### 3.2. Effects of Coating Material on Measured Bond Strength and Coating Thickness

The effect of coating type on the measured tensile bond strength after 72 h of water curing is depicted in Figure 12. The bond strength values for S1, S2, and S4 were 7.67, 6.95, and 5.71 MPa, respectively, indicating better adhesion performance than those of coating materials S3, S5, and S6. The lowest bond strength value was 0.26 MPa when S5 was applied using EQ3. The average bond strength for S6 was 1.25 MPa, which was the lowest adhesion performance. S1 and S2 exhibited significantly high bond strengths with EQ1. The results show that coating thickness greatly varied depends on the type of coating equipment rather than the type of coating material. Figure 13 summarizes the measured coating thickness values after 72 h of curing. With the exception of EQ4, the coating thicknesses of S5 and S6 were observed to be relatively greater than those of the other coatings. The coating thickness of S1 was measured using different coating equipment, and it was confirmed that the average coating thicknesses with using EQ4 was 0.04 mm (with SD 0.02), which is in the range of 0.01–0.06 mm. On the other hand, when the coating material was applied using EQ3, the average coating thickness was 0.48 mm (with SD 0.28), which is in the wide range of 0.09–0.75 mm, showing the unstable thickness values.

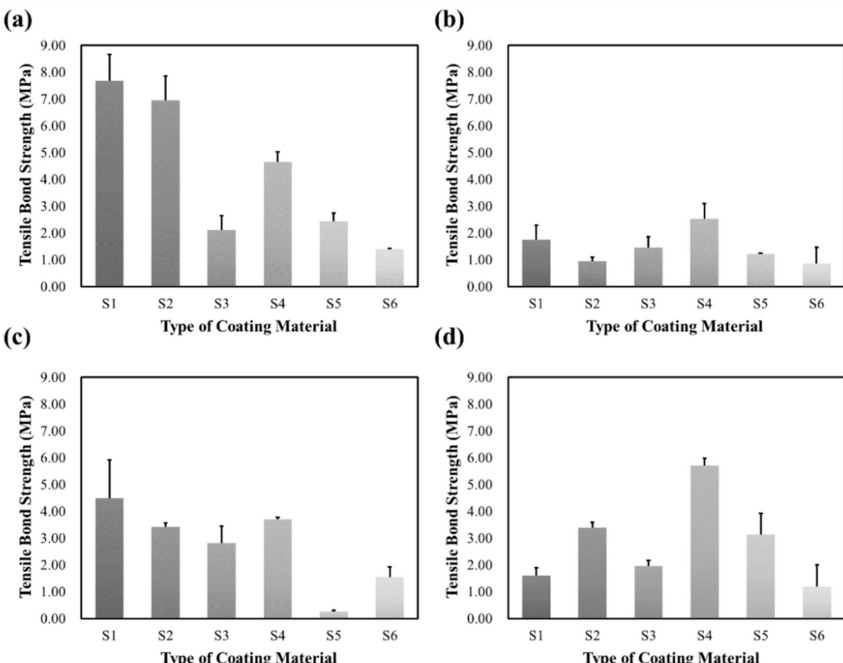

**Figure 12.** Averaged bond strength values at 72 h curing ((**a**) EQ1; (**b**) EQ2; (**c**) EQ3; (**d**) EQ4).

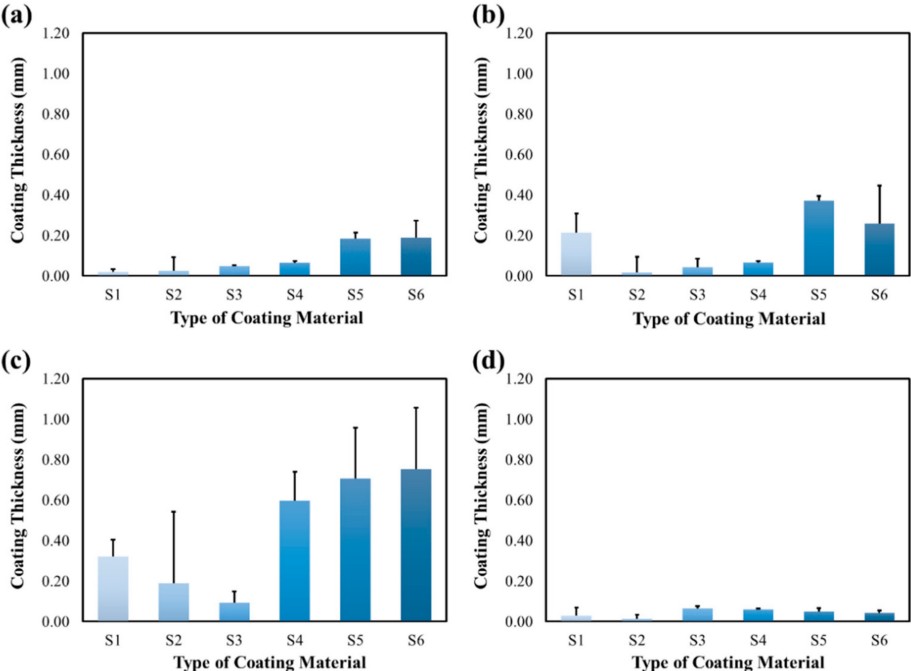

**Figure 13.** Averaged coating thickness values at 72 h curing ((**a**) EQ1; (**b**) EQ2; (**c**) EQ3; (**d**) EQ4).

Based on these results, the type of coating material applied does not appear to affect the bond strength significantly. Coating materials S1, S2, and S4 showed good adhesion performance, and it was confirmed that the adhesion performance of S6 was relatively poor compared to the others. These results may be attributed to various causes such as the dissolution of the coating agent, human error, and conditions of the working environment. In addition, because the curing time of 72 h investigated in this study is relatively short, further experiments need to be conducted by applying longer curing times so that sufficient curing can be achieved. Furthermore, the coating thickness is apparently more affected by the coating equipment rather than the type of coating material. Therefore, to analyze the difference in coating thickness according to the type of coating material, it is necessary to eliminate factors other than variable factors, especially external factors such as human error, by setting the coating amounts and coating areas to be similar.

*3.3. Effects of Curing Time on Measured Bond Strength and Coating Thickness*

Figures 14 and 15 depict the effects of the differences in curing times on the tensile bond strengths and coating thicknesses. Initially, it was expected that the bond strength would improve with increasing curing time, on the basis of a previous study [26]; however, the effects of underwater curing times on the adhesion strengths were not significant. In Figure 14b, some of the coating materials, such as S2, were cured for 72 h, but they exhibited lower bond strengths than those measured after 24 and 48 h of curing. Furthermore, the adhesion of the coatings and their work performance in the underwater condition are both less than those for the conventional atmosphere condition, and it is difficult to apply a uniformly thick coating.

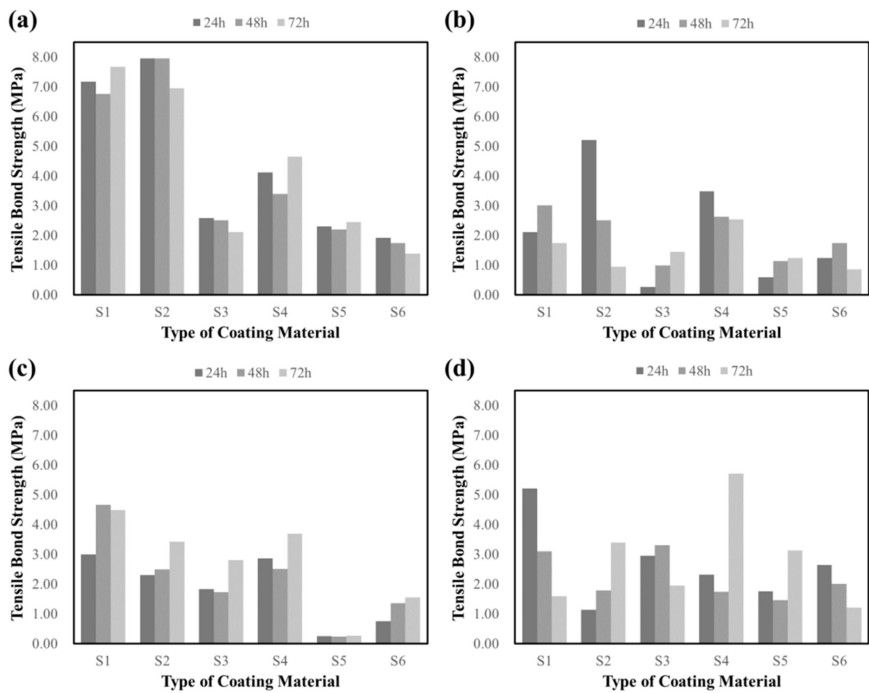

**Figure 14.** Effect of curing time on measured tensile bond strength ((**a**) EQ1; (**b**) EQ2; (**c**) EQ3; (**d**) EQ4).

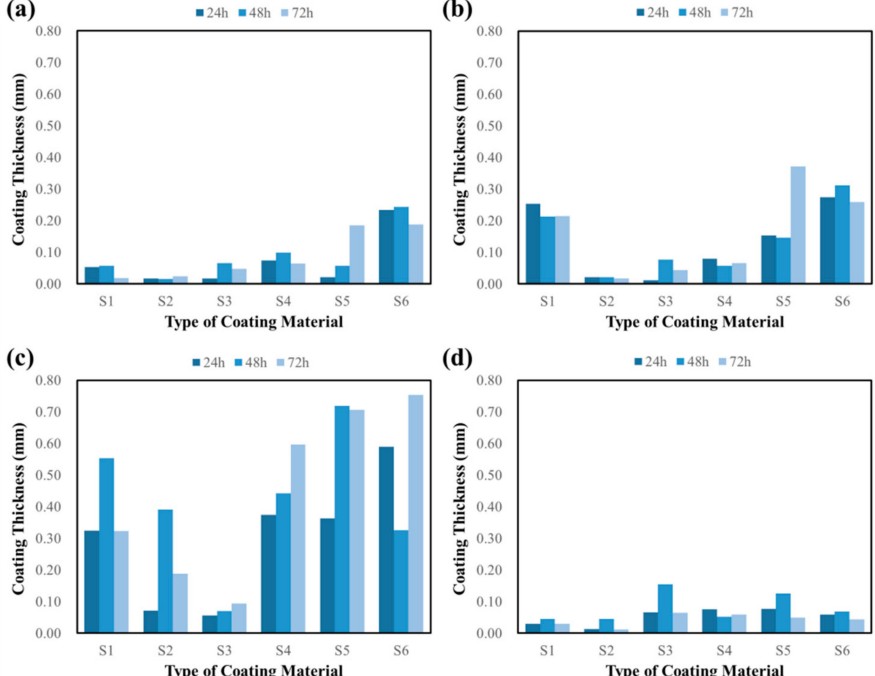

**Figure 15.** Effect of curing time on measured coating thickness ((**a**) EQ1; (**b**) EQ2; (**c**) EQ3; (**d**) EQ4).

In summary, the effects of the different curing times on measured bond strength values were not significant, and further studies and experiments with a curing time longer than 72 h are necessary. Although the coating materials were applied on the surfaces of the three specimens at the same time by the same operator, the coating thickness was greatly varied in the case of EQ3. This variation can be attributed to the high density and material stiffness of the brush of EQ3 compared to the brushes of the other equipment.

### 3.4. Relationship between Coating Thickness and Bond Strength

　　　Figure 16a–f show the relationships between the measured coating thickness and tensile bond strength values of each coating. Figure 16c–f show the relationships between the measured tensile bond strengths and coating thicknesses for each coating equipment. It can be seen that EQ1 has a relationship between these two variables, with the value of $R^2$ being 0.70, whereas the other equipment did not manifest any relationships at this time. This is because EQ1 has positive influences on both the adequate distribution of the applied coating and earlier hardening compared to the others. Other coating equipment developed in this study should be further improved to contribute toward better bonding while achieving low material loss in underwater condition. Higher bond strengths were expected upon increasing the coating thicknesses because of breakage in the coatings rather than breakage in the substrates or interfaces. However, the bond strengths tended to decrease slightly as the film thickness increased. Thus, increasing the coating thickness did not increase the bond strength because of incomplete curing of the thick coating owing to the relatively short curing.

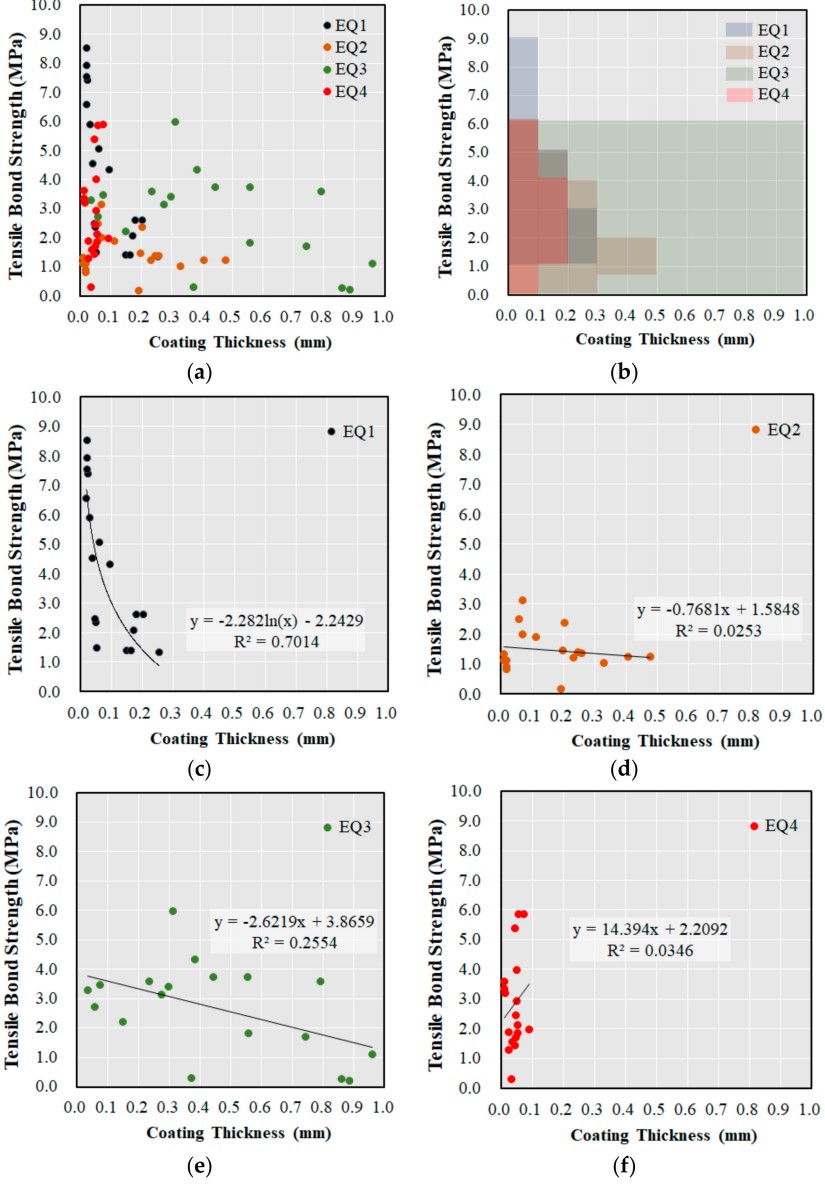

**Figure 16.** Correlations between tensile bond strength and coating thickness values measured at 72 h. (**a**) overall relationship; (**b**) overall pattern; (**c**) EQ1; (**d**) EQ2; (**e**) EQ3; (**f**) EQ4.

In other words, because the curing time in the underwater condition was limited to only 72 h, the bond strength of the relatively thick coating films (1.0 mm thick or more) was reduced. Further research is required to clarify the relation between coating thickness and tensile bond strength.

### 3.5. Microstructure of the Interfacial Zone between the Substrate and Representative Epoxy Coatings

Figure 17a–d show the SEM images representing the microstructures of the interfaces between the substrates and representative epoxy coatings. As seen in Figure 17a,b, the interfaces between the substrates and coatings remain intact without any visible cracks or separations at the interfaces. On the other hand, S4 and S5 coatings exhibited residual voids or separations at the interface, as seen in Figure 17c,d. These trends are consistent with the measured tensile bond strength values described in Figure 12a. Specifically, the measured bond strength values for S4 and S5 coatings were equal to 47% of those of the S1 and S2 coatings. Thus, it is concluded that the type of epoxy coating can also influence the interfacial bond property.

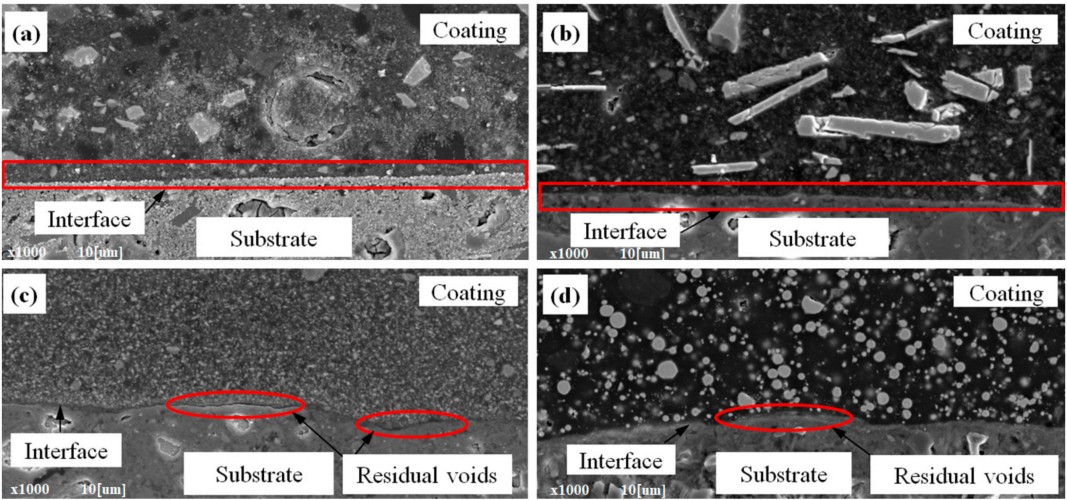

**Figure 17.** SEM images of the interfacial zone between substrate and representative epoxy resin coatings ((**a**) S1; (**b**) S2; (**c**) S4; (**d**) S5).

## 4. Conclusions

In this study, experiments were conducted to develop coating equipment for the maintenance of submerged structures and to investigate the bonding performance of epoxy coatings using the developed equipment for underwater application. The performance was evaluated by applying six different epoxy coatings to the surfaces of specimens in the underwater condition using two types of pre-existing and two developed equipment. The tensile bond strength and coating thickness were selected as performance indices, and three variables, namely, coating equipment, coating material, and curing time, were considered. The relationship between coating thickness and tensile bond strength of the coating material was analyzed by considering the experimental results according to the equipment and variables. The conclusions derived from the experimental results are as follows:

1. The difference in bonding performance can be attributed to the type of coating equipment rather than the coating itself, and the pre-existing equipment EQ1 showed the best performance for bonding.
2. For the same generic type of coating, the bond performance and coating thickness vary between manufacturers, and these results are consistent with those of previous research.
3. Within the range of the curing time considered in this study, i.e., 72 h, there was no significant effect on measured tensile bond strength over time. This may be due to the fact that the maximum curing time of 72 h was insufficient; thus, underwater curing time should be further increased.

4. It is necessary to develop appropriate surface-treatment brushes and improve equipment workability by reducing the weights of the equipment and adjusting the positions of the injection valves.

5. The relationship between coating thickness and tensile bond strength of the coating material was varied according to the type of coating equipment used. For EQ1, EQ2, and EQ4, the thicknesses of the coating films were evenly distributed within 0.5 mm, but in the case of EQ3, the coating-film thickness was distributed unevenly. EQ1 showed a notable trend between the measured strength and thickness values, while the other equipment did not have exhibit any similar relationships.

Finally, to solve the problems summarized in this study, additional underwater experiments must be performed by simulating the marine environment and considering longer curing times. In addition, it is necessary to incorporate the advantages of the surface-treatment mechanisms of pre-existing equipment in the next generation of coating equipment and to improve the issues noted for the developed equipment, such as large mass. Further experiments using improved coating equipment are expected to provide more accurate results for both the durability of the coated structure in the underwater environment and the relationship between adhesion properties (tensile bond strength and coating thickness). It should also be noted that this study is part of an ongoing research project. In future studies, the exotherms of different types of resins should be investigated for optimum underwater coating.

**Author Contributions:** Conceptualization, M.O.K.; methodology, M.O.K.; validation, S.K., H.H. and M.O.K.; investigation, S.K.; data curation, H.H.; writing—original draft preparation, S.K.; writing—review and editing, M.O.K.; visualization, S.K.; supervision, M.O.K.; project administration, T.H.H.; funding acquisition, T.H.H.

**Funding:** This research work was supported by the "Development of application technologies for ocean energy and harbor and offshore structures (KIOST Project Number: PE99731)".

**Conflicts of Interest:** The authors declare no conflict of interest.

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
