# Peer review of "Early-Age Tensile Bond Characteristics of Epoxy Coatings for Underwater Applications"

_coatings, doi:10.3390/coatings9110757_

Round 1

Reviewer 1 Report

The manuscript titled "Early-Age Tensile Bond Characteristics of Epoxy Coating for Underwater Application" discusses the tensile bond strength and the coating thickness were analyzed according to the type of equipment, type of coating, and curing time for underwater applications. The results are interesting considering the application of coatings in underwater application.

Reviewer recommend to add few details should be added in the final version

The authors should discuss the details of different types of epoxy resin and hardener used. The authors should mention whether they cross-checked the exotherms of the different resins to find out the optimum curing temperature and time.

Reviewer 2 Report

The paper “ Early-Age Tensile Bond Characteristics of Epoxy 2 Coating for Underwater Application” presents an interesting experimental work made to evaluate if the coating equipment affects the quality of the coating of 6 different commercial formulations for underwater application. It contains valuable technological information. For example the relationship between the pot life and the working temperature for different commercial formulations.

The authors investigated the tensile bond strength and the coating thickness were analyzed according to the type of equipment, type of coating, and curing time.

In my opinion, this paper deserves publication however some comments should be addressed before in order to improve the manuscript and the data discussion:

Point 1 please check the way the figures are numbered.

They are in order: figure 1, 2, 3, 4, 5, 6, 7, 8, 9, and suddenly figure 70, figure 81, figure 92, then 103, 114, and Figure 125

Point 2

The authors namely S1, S2, S3..., S6 the different commercial formulation, but if they would prefer not to say the commercial name of the used formulation they should at least indicate the main component of it. For instance, if it is a bifunctional DGEBA, or a tetrafunctional TGMDA or any other epoxy resin, the type of the hardener for instance and aromatic, or aliphatic amine, the presence of diluent in order to understand in what the formulation differs from a chemical point of view.

Point 3)

The author made a very big effort to prepare different samples and to analyze them, but the relation between the used methodologies and the performance of the coating is not straightforward. For instance, it is very difficult to get valuable information from figure 125 “Relation between measured bond strength and coating thickness on steel specimen”.

I believe the bond stretch also depend on the morphology of the sample, and I strongly suggest adding some SEM (electronic microscopy) analysis to improve the quality of the paper.

Author Response

Point 1: Please check the way the figures are numbered. They are in order: figure 1, 2, 3, 4, 5, 6, 7, 8, 9, and suddenly figure 70, figure 81, figure 92, then 103, 114, and Figure 125.

Response 1: All figure numbers were corrected as the reviewer commented.

Point 2: The authors namely S1, S2, S3..., S6 the different commercial formulation, but if they would prefer not to say the commercial name of the used formulation they should at least indicate the main component of it. For instance, if it is a bifunctional DGEBA, or a tetrafunctional TGMDA or any other epoxy resin, the type of the hardener for instance and aromatic, or aliphatic amine, the presence of diluent in order to understand in what the formulation differs from a chemical point of view.

Response 2: To reflect the comment received, Table 1 was corrected to include the details (product name and company) of epoxy coatings used. We also contacted with the company, however, they do not provide further information such as main component and type of resin, hardener.

Table 1. Material properties of selected coatings.

Coating material

(code)

Tensile bond strength

(MPa)

Density

(g/cm3)

Working time

(min)

Product name

  Company

S1

12.7

1.75

45 @ 20°C

5831

Belzona

S2

16.6

1.60±0.1

30 @ 30°C

RS500P

Chemco

S3

12.0

1.60

30 @ 25°C

M500

Westsea

S4

6.9

1.55

45∼60 @ 20°C

28.15

A&E Systems

S5

17.0

1.82

45∼60 @ 20°C

28.14

A&E Systems

S6

Non available

40 @ 23°C

5027

AkzoNobel

Point 3: The author made a very big effort to prepare different samples and to analyze them, but the relation between the used methodologies and the performance of the coating is not straightforward. For instance, it is very difficult to get valuable information from figure 125 “Relation between measured bond strength and coating thickness on steel specimen”.

Response 3: To reflect the comment received, figure 15 was corrected to express the effect of developed coating equipment. And the following sentences were added in the section of 3.4. Relation between Coating Thickness and Bond Strength (Page 12, Line 307-Page 13, Line 316)

“Figure 15 (a) to (f) shows the relationship between the measured coating thickness and tensile bond strength values of each coating. Specifically, the effects of each coating equipment on each measurement were further described in Figure 15 (c) to (f). As shown in Figure 15(b), the coating applied to the steel specimens with using three coating equipment (EQ1, EQ2, and EQ4) resulted in a bond-strength distribution of up to 9.0 MPa within the thickness of less than 0.5 mm. In the case of EQ3, however, no constant trend was observed between the measured coating thickness and the tensile bond strength.”

Point 4: I believe the bond stretch also depend on the morphology of the sample, and I strongly suggest adding some SEM (electronic microscopy) analysis to improve the quality of the paper

Response 4: The authors understand what the reviewer would like to say and strongly agree with received comment. However, test result reported herein is from our previous research work and it is not available to conduct SEM analysis at this moment. We have a research plan to conduct the experimental study with using the same type of epoxy coating and we will report SEM results in our future study. Once again, thank you for your valuable comments and suggestions.

Round 2

Reviewer 2 Report

dear Author,

I read carefully your comments. Unfortunately, I dont' believe the work without a morphological study (SEM or any other techniques) is complete. The relation between Coating Thickness and Bond Strength is not commented at all and no scientific justification on the obtained results are provided. The answer <<However, test result reported herein is from our previous research work and it is not available to conduct SEM analysis at this moment>> is not a justification to avoid to perform editing to improve the paper and makes it publishable.

Although quite a lot of work and scientific tests is here described it has not scientific significance. I do believe authors should put a little bit of effort to improve it before consider it publishable.

Round 3

Reviewer 2 Report

the points raised by the reviewer have been satisfactorily addressed, the paper now is suitable for the publication. 

Author Response

Dear Reviewer,

Thank you for your time and consideration.